# Antiparkinsonian Agents in Investigational Polymeric Micro- and Nano-Systems

**DOI:** 10.3390/pharmaceutics15010013

**Published:** 2022-12-20

**Authors:** Nicola Paccione, Mahdieh Rahmani, Emilia Barcia, Sofía Negro

**Affiliations:** 1Department of Pharmaceutics and Food Technology, School of Pharmacy, Universidad Complutense de Madrid, Ciudad Universitaria s/n, 28040 Madrid, Spain; 2Institute of Industrial Pharmacy, Universidad Complutense de Madrid, Ciudad Universitaria s/n, 28040 Madrid, Spain

**Keywords:** Parkinson’s disease, polymeric microparticles, polymeric nanoparticles, neurotrophic factors, monoamine oxidase inhibitors, antioxidants, dopamine agonists, catechol-o-methyl transferase inhibitors, alpha-synuclein-targeting agents, anti-inflammatory agents

## Abstract

Parkinson’s disease (PD) is a devastating neurodegenerative disease characterized by progressive destruction of dopaminergic tissue in the central nervous system (CNS). To date, there is no cure for the disease, with current pharmacological treatments aimed at controlling the symptoms. Therefore, there is an unmet need for new treatments for PD. In addition to new therapeutic options, there exists the need for improved efficiency of the existing ones, as many agents have difficulties in crossing the blood–brain barrier (BBB) to achieve therapeutic levels in the CNS or exhibit inappropriate pharmacokinetic profiles, thereby limiting their clinical benefits. To overcome these limitations, an interesting approach is the use of drug delivery systems, such as polymeric microparticles (MPs) and nanoparticles (NPs) that allow for the controlled release of the active ingredients targeting to the desired site of action, increasing the bioavailability and efficacy of treatments, as well as reducing the number of administrations and adverse effects. Here we review the polymeric micro- and nano-systems under investigation as potential new therapies for PD.

## 1. Introduction

Parkinson’s disease (PD) is the second most prevalent neurodegenerative disease worldwide [1], affecting 6.1 million people in 2016 [2] and rising to 8.5 million in 2019 [3]. This devastating disease is characterized by progressive destruction of dopaminergic tissue (DT) in the central nervous system (CNS), predominantly in the substantia nigra pars compacta (SNpc), with the formation of alpha-synuclein (α-syn) protein aggregates, also known as Lewy bodies [4]. It is a progressive and chronic process that usually initiates with non-motor prodromal symptoms (olfactory loss, depression, constipation, or sleep disorders) leading, as the neurodegeneration progresses to the appearance of its characteristic motor symptoms (rigidity, tremor, bradykinesia, postural alterations, etc.) [5,6]. Finally, in the advanced stages of the disease, controlling the motor symptoms becomes very complicated, and more non-motor symptoms appear (urinary problems, pain, dementia, fatigue, etc.) that severely compromise the quality of life of PD patients [7,8].

The exact cause of PD is not yet fully understood. Only 5–10% of the cases are attributable exclusively to genetic alterations, with most of the cases being related to a combination of both genetic and environmental factors [9]. The main pathological feature of PD is the loss of dopaminergic neurons in the substantia nigra (SN), associated with abnormal protein aggregation [10].

There are several therapeutic options for the management of PD, including the administration of levodopa, dopamine agonists, monoamine oxidase B (MAO-B) inhibitors, catechol-o-methyl transferase (COMT) inhibitors, and anticholinergic agents, among others [8,11,12] (Figure 1). 

However, to date there is no cure for the disease [8]. Furthermore, although current therapies for PD ameliorate the symptoms in the early phases of the process, they are less effective as the neurodegeneration advances [13]. This occurs with levodopa, the endogenous precursor of dopamine (DA), which is considered the “gold standard” for PD and commonly used as a DA replacement agent, being still the most efficacious drug to treat the symptoms of PD [14,15].

Today, levodopa-based therapies continue to be the standard care for PD, as they are effective in treating the disease’s motor symptoms and tend to be of relatively low cost. However, although these medications offer valuable symptomatic relief, as neurodegeneration advances, their use is often associated with significant and sometimes intolerable side effects. In addition to new therapeutic options, there is a need for improved efficiency of the existing ones, as many agents have difficulties in crossing the blood–brain barrier (BBB) to achieve therapeutic levels in the CNS or exhibit inappropriate pharmacokinetic profiles, thereby limiting their clinical benefits (Figure 2).

To overcome these limitations, an interesting approach is the use of drug delivery systems, such as polymeric microparticles (MPs) and nanoparticles (NPs) [16]. These systems allow for controlled release of the active ingredients, targeting the desired site of action, increasing the bioavailability and efficacy of treatments, as well as reducing the number of administrations and adverse effects [17,18] (Figure 3).

Within the micro- and nano-systems approved for clinical use, the largest group is that of polymeric particles [19,20]. Among the polymers most commonly used when developing MPs and NPs are polylactide-co-glycolic acid (PLGA) [21], polycaprolactone (PCL) [22], chitosan [23], poly (ethylene glycol) (PEG) [24], and poly (butylcyanoacrylate) (PBCA) [25], due to their adequate characteristics. 

The aim of the present study is to address the challenges of routine traditional pharmacotherapy of PD and to review the antiparkinsonian agents encapsulated within polymeric micro- and nano-systems, which are under investigation as potential new strategies for the treatment of PD.

## 2. Investigational Polymeric Microparticles for the Treatment of Parkinson’s Disease

In recent years, different multiparticulate systems, such as polymeric MPs, have been designed as potential therapeutic approaches for PD. MPs are structures with sizes ranging from 1 to 1000 µm [26]. Among the advantages of MPs is the possibility of achieving the controlled release of the active ingredients at the target site. This is of great interest for drugs that have limited access to the CNS. In addition, reduced adverse systemic effects and less frequent dosing intervals can be also achieved by MPs. Under investigation are different antiparkinsonian agents encapsulated within polymeric MPs, including neurotrophic factors such as glial-cell-derived neurotrophic factor (GDNF) and vascular endothelial growth factor (VEGF), antioxidant agents, inhibitors of different enzymes such as monoamine oxidase (MAO) and catechol-O-methyltransferase (COMT), anti-inflammatory compounds, as well as α-synuclein-targeting agents. 

Categories of active ingredients encapsulated within polymeric MPs for the treatment of PD include the following.

### 2.1. Neurotrophic Factors

Neurotrophic factors (NTFs) are biological molecules that influence several neuronal functions, including cell survival and axonal growth [27]. For instance, GDNF has demonstrated to be able to protect and promote neuronal growth, also improving motor functions in animal models of PD [27]. However, as the BBB is a major entry restriction, its high molecular weight and water solubility hinders the access of GDNF to the brain parenchyma. Thus, one viable route of administration is by means of viral vectors integrating the GDNF gene, or by implanting cells, semipermeable structures, or grafts that secrete the neurotrophic factor [28]. In addition, GDNF exhibits a short biological elimination half-life due to its unstable nature. For these reasons, encapsulating this neurotrophic factor within polymeric microparticulate systems may be an interesting approach for brain delivery, as these systems can protect GDNF against external factors, also providing sustained release [29]. 

The use of GDNF has been Investigated in combination with cell transplants, although this strategy does not lead to direct cell growth, which is necessary for full functional recovery. Furthermore, GDNF must be released transiently at low doses rather than in a constant manner over several months, since continuous release may affect synaptic integration of the transplanted and host tissues. Therefore, given the complex mechanism associated with PD, an interesting strategy may be the design of devices able to release GDNF in the nigral region to improve immediate transplant survival, and in addition, possibly release another factor, such as BDNF, to stimulate neurite outgrowth in the striatal region [30]. 

Brain-derived neurotrophic factor (BDNF) stimulates and controls growth of new neurons from neural stem cells (neurogenesis) [31,32,33]. In vitro studies have also provided evidence that BDNF can improve DA production, axonal extension, and survival in cultured dopaminergic neurons [34]. Abnormally low levels of this factor have been associated with neurodegenerative diseases such as PD [35]. BDNF could be used to reduce neurodegeneration and enhance the success of cell transplants required for long-term therapy and potential reversal of disease by promoting reinnervation [36]. However, clinical efficacy has been severely limited by its limited access to the CNS [28]. 

In human traumatic brain injury (TBI), day-of-injury serum BDNF has been associated with TBI diagnosis also providing 6-month prognostic information regarding recovery from the injury [37].

Vascular endothelial growth factor (VEGF) is an angiogenic factor with specificity for endothelial cells that has demonstrated neuroprotective effects in animal models of PD [38]. However, the use of VEGF as a neuroprotective factor is complicated, as its effect is dose-dependent, with high levels of VEGF inducing brain edema [39]. As with other NFs, clinical use is limited by its rapid degradation and difficulty in crossing the BBB [40]. To obtain a continuous and direct release of VEGF to the CNS, different intracranial administration strategies, such as MP administration, have been tested in animal models of PD [41] (Figure 4).

Therefore, NFs (GDNF, BDNF, VEGF) are of interest as potential therapeutic strategies for PD. Different polymeric systems have been developed that include GDNF or VEGF alone [40,42], GDNF and BDNF together [43], or VEGF [40]. 

For instance, Gujral et al. [44] developed polymeric PLGA/collagen MPs by a water-in-oil-in-water (W/O/W) double emulsion method in which GDNF was fused with collagen-binding peptide (CBP) and immobilized to the inner collagen phase. The controlled release of GDNF-CBP was obtained when compared with standard MPs, with the system being non-cytotoxic up to concentrations of 300 μg/2 × 10^5^ cells in neural stem/progenitor cells (NSPCs). In addition, culture of MPs with NSPCs cells induced differentiation into mature neurons, with the bioactivity of released GDNF being like that of recombinant human GDFN (rhGDNF). For this, GDNF-CBP MPs could be useful for the management of PD and other neurodegenerative diseases. 

Jollivet et al. [42] prepared GDNF MPs that released in vivo the neurotrophic factor for 2 months. rhGDNF-loaded MPs were elaborated using PLGA as a biodegradable polymer and were implanted into the brains of parkinsonian rats treated with 6-hydroxydopamine (6-OHDA) to induce a partial progressive and retrograde lesion of the nigrostriatal system. In this study, the MPs were well tolerated and induced sprouting of the preserved dopaminergic fibers with synaptogenesis. Other authors [45] have also demonstrated that GDNF-releasing MPs were able to protect dopaminergic neurons for 6 weeks when concomitantly injected with 6-OHDA by stereotaxic procedure. 

Garbayo et al. [46] prepared N-glycosylated recombinant GDNF-loaded PLGA MPs. This delivery system was tested in an animal model of PD in which a unilateral lesion was induced in the nigrostriatal area by stereotaxic administration of 6-OHDA every 2 weeks. The amphetamine-induced rotational asymmetry test was conducted in the animals. The formulation developed was able to release the active ingredient for 5 weeks, improving the rotational behavior induced by amphetamine in the animals treated with GDNF as well as increasing the density of TH-positive fibers at the striatal level, without causing toxicity. The same authors [47] also prepared GDNF-loaded PLGA MPs, which were evaluated in macaques (*Macaca fascicularis*) after inducing PD-like symptoms with 1-methyl-4-phenyl-1,2,3,6-tetrahydropyridine (MPTP). The MPTP model has some advantages over 6-OHDA as it does not require stereotaxic surgery for its administration [48,49]. The MPs were administered unilaterally into the putamen of parkinsonian monkeys with severe nigrostriatal degeneration. After 9 months, administration of microencapsulated GDNF (single dose, 25 mg of GDNF) led to sustained GDNF levels in the brain, resulting in motor enhancement and restoration of dopaminergic function. This was reflected by a bilateral increase in the density of striatal dopaminergic neurons. In addition, GDNF was retrogradely transported to the substantia nigra, bilaterally increasing the number of dopaminergic and total neurons, regardless of the degeneration produced. In addition, the administration of GDNF-loaded PLGA MPs was found to be safe, not causing immunogenicity, cerebellar degeneration, or weight loss. Therefore, the safety and efficacy shown by this new therapeutic system may represent an important basis for the clinical development of GDNF MPs [47]. 

Lampe et al. [43] developed a new delivery system consisting of PLGA MPs entrapped within degradable PEG-based hydrogel devices to locally release GDNF and BDNF with two different release profiles. The authors aimed for locally deliver GDNF or BDNF to brain areas associated with PD. GDNF was included to improve immediate transplant cell survival [30] and BDNF to stimulate neurite outgrowth in the striatal region. In the experiments carried out in Sprague Dawley rats, stereotaxic implantation of the hydrogels was conducted with the BDNF end oriented to the striatum and the GDNF end to the substantia nigra, using minimally invasive single penetration for implantation. Doses of NTFs encapsulated within each hydrogel were 1 ng for BDNF and 10 ng for GDNF. The device delivered the NTFs in a region localized within 100 µm of the bridge, but not exactly to the targeted rostral or caudal ends. BDNF was slowly delivered over a period of 56 days, while a bolus of GDNF was released at around 28 days. The timed delivery of NTFs from the implanted devices using a multifaceted PEG hydrogel/PLGA microparticle strategy markedly reduced the microglial response relative to sham surgeries, which may constitute an interesting approach for the potential treatment of neurodegenerative diseases, such as PD. 

The effect of VEGF, GDNF, and their combination was evaluated by Herrán et al. [40]. These authors developed PLGA MPs loaded with GDNF and/or VEGF (GDNF-MPs and VEGF-MPs, respectively) which were administered to a 6-OHDA PD model induced in Sprague Dawley rats. Animals treated with at least one of the NTFs showed regenerative improvement in the amphetamine-induced turnover tests with respect to control animals, with the most effective intervention being achieved by GDNF-loaded MPs. In addition, recovery of the neuronal tissue was quantified at the end of the treatment by means of optical densitometry and immunohistochemical tests. Non-statistically significant differences were found, although animals treated with GDNF-loaded MPs or VEGF/GDNF-loaded MPs showed greater recovery in the injured area than those treated with VEGF-MPs or blank MPs. Both GDNF individually and the combination of GDNF and VEGF encapsulated within polymeric MPs exhibited regenerative effects in this experimental model of PD.

MPs prepared with biodegradable and biocompatible polymers represent a potential approach for the delivery of NFTs to the CNS; however, in most of the experiments carried out with NFT-loaded MPs, either the PD animal model assayed was that of 6-OHDA, which involves stereotaxic administration, thereby facilitating penetration of the MPs, or MPs were surgically implanted. As MPs cannot cross the BBB, less invasive administration techniques should be investigated to adequately establish the therapeutic potential of polymeric NFT-loaded MPs in neurodegenerative diseases. 

### 2.2. MAO Inhibitors/Antioxidants

Oxidative stress is a result of various metabolic activities which are essential for life and usually leads to the formation of reactive oxygen species (ROS) and reactive nitrogen species (RNS). It has been associated with the development of PD [50], as increased oxidative stress has been related to the overexpression of α-syn aggregates [51].

The mechanisms involved in neuronal degeneration occurring in PD are complex and remain to be fully elucidated, although it is known that the loss of dopaminergic neurons in the substantia nigra pars compacta (SNpc) is responsible for the characteristic motor symptoms present in the disease [1,2]. Accumulating evidence suggests that oxidative damage and mitochondrial dysfunction contribute to the cascade of events leading to degeneration of dopaminergic neurons [52,53,54]. This is supported by post-mortem brain analyses showing increased levels of 4-hydroxyl-2-nonenal (HNE), a by-product of lipid peroxidation [55], the formation of DNA and RNA oxidation products (8-hydroxy-deoxyguanosine and 8-hydroxy-guanosine) [56,57], and carbonyl modifications of soluble proteins [58]. The link between oxidative stress and dopaminergic neuronal degeneration is further supported by modelling the motor aspects of PD in animal models using toxins that cause oxidative stress, including MPTP, rotenone (RT), 6-OHDA, and 1,1′-dimethyl-4,4′-bipyridinium dichloride (paraquat) [59,60,61]. For this reason and given that oxidative imbalance in PD has a multifactorial origin, the use of antioxidant agents could be a viable therapeutic strategy [62]. In this regard, multiple alternatives have been proposed based on the use of iron chelators, ROS scavengers (vitamins, polyphenols, glutathione, etc.), and other substances (vegetable extracts, melatonin, nicotine, etc.) [62,63].

In addition, it seems that MAO-B (monoamine oxidase-B) plays an important role in the production of reactive oxygen intermediates (ROI) in activated astrocytes. MAO-B metabolizes the MPTP toxin to 1-methyl-4-phenylpyridium (MPP+), leading to the production of ROI and eventually to cell death [64]. 

Rasagiline (RG) is a therapeutic agent belonging to the group of MAO-B inhibitors (MAOIs) which has demonstrated neuroprotective activity against MPTP and 6-OHDA animal models of PD [65,66]. Rasagiline is used for the symptomatic treatment of PD. A recent study conducted in PD patients by Im et al. [67], when investigating the effects of RG on regional cerebral blood flow (rCBF) by single-photon-emission-computed tomography (SPECT), showed that adjunctive RG therapy has beneficial effects on perfusion in the precuneus of PD patients due to its neuroprotective effects.

The low oral bioavailability of RG (around 36%) and its short elimination half-life (0.6–2 h) [68] make it a suitable candidate for the design of controlled-release systems. For this, Fernández et al. [69] developed PLGA MPs loaded with RG mesylate (RM). The efficacy of this system was evaluated in male Wistar rats in an RT-induced model of PD. Daily intraperitoneal (i.p.) administration of RT at a dose of 2 mg/kg/day resulted in neuronal degeneration and behavioral deficits resembling those occurring in PD. Treated animals received the same daily dose of RT for 45 days and RM in saline (1 mg/kg/day) or RM-loaded PLGA MPs, which were assayed at two dose levels: high dose (amount of MPs equivalent to 15 mg/kg RM injected every 15 days) and low dose (amount of MPs equivalent to 7.5 mg/kg RM injected every 15 days). The results demonstrated a robust effect of high-dose RM-loaded MPs on all behavioral tests (catalepsy, akinesia, swim test) which resulted in better outcomes than RM given in solution (1 mg/kg/day). Furthermore, Nissl staining of brain sections showed selective degeneration of the substantia nigra (SN) dopaminergic neurons in animals treated with RT, which was markedly reverted by the administration of high-dose RM-loaded PLGA MPs. Interestingly, PET/CT analysis (positron emission tomography/computed tomography) using 18F-DG (fluorodeoxyglucose F18) resulted in mean increases in the radiotracer in striatum and SN of around 40% when RM-loaded PLGA MPs were given to the animals, thereby indicating the efficacy of the microparticulate drug delivery system developed for RM. 

This study continued in 2012 [70]. In this case, an advanced stage of neurodegeneration was achieved by daily i.p. injections of RT (2 mg/kg). On day 15, animals received RM in saline (1 mg/kg/day) or encapsulated within PLGA microspheres (amount of microspheres equivalent to 15 mg/kg RM given on days 15 and 30). After 45 days RM showed a robust effect on all outcomes evaluated (behavioral tests, Nissl staining, and PCR or Polymerase Chain Reaction), with non-statistically significant differences found between its administration in solution or encapsulated within MPs; however, with the new delivery system, administration of RM could be performed every two weeks instead of daily. 

Kanwar et al. [22] evaluated the efficacy of RM-loaded polycaprolactone (PCL) MPs in Sprague Dawley rats. For the development of PD-like symptoms, stereotaxic infusion of RT (6 μg/2 μL vehicle) was conducted in the animals. After recovery from surgery, animals received RM in solution (1 mg/kg/day), blank PCL MPs, or RM-loaded PCL MPs (equivalent to 30 mg/kg RM given once a month by subcutaneous (s.c.) injection). Significant differences in behavioral tests (locomotor activity, grip strength) and biochemical markers of oxidative stress (lipid peroxidation, reduced glutathione, etc.) were observed between the RM-treated groups and control animals. Non-statistically significant differences were found when RM was given in solution or encapsulated within PCL MPs, but administration of the microparticulate formulation could reduce the need for frequent dosing intervals, thereby resulting in better patient compliance in the treatment of PD.

A few polymeric microparticulate systems have been developed for the encapsulation of MAO-B inhibitors. From the formulation developed, satisfactory results were obtained in animal models of PD, although several issues should be addressed, such as pharmacokinetics and toxicity, as long-term systemic medication commonly leads to deleterious side effects.

### 2.3. Dopamine Agonists

DA receptor agonists may be able to prevent the nigrostriatal dopaminergic cell loss occurring in PD due to their antioxidant and levodopa-sparing effects. For this, pramipexole [71,72,73], apomorphine [74], and ropinirole [75] have been studied in different neurodegenerative animal models, with the outcomes of these studies demonstrating that, in general, these DA agonists are able to prevent the loss of dopaminergic neurons. 

Furthermore, recent therapeutical approaches for PD are exploring the use of radical scavenging materials, as polymers as catechol structures have demonstrated antioxidant activities due to their physicochemical properties. In this regard, Newland et al. [76] synthesized photocrosslinkable DA-containing poly(β-amino ester) (DPAE) from poly(ethylene glycol) diacrylate (PEGDA) and dopamine hydrochloride using Michael-type addition. The authors developed a water-in-oil emulsion technique to photocrosslink the polymer into spherical MPs. The ability of the microparticulate formulation to capture ROS was analyzed by the 2,2-diphenyl-1-picrylhydrazyl (DPPH) method. DPAE MPs at concentrations of 5250 spheres/mL and 7000 spheres/mL reduced the formation of DPPH radicals up to 47 ± 9% and 56 ± 5%, respectively. In contrast, PEG MPs at a concentration of 7000 spheres/mL led to a reduction of only 5 ± 1%. The free radical activity of DPAE MPs increased in a dose-dependent manner up to 56%, with this level of antioxidant activity being slightly higher than that produced by ascorbic acid (10 μM), a well-known antioxidant. Furthermore, tests conducted in dopaminergic SH-SY5Y cells, primary astrocytes, and primary embryonic rat ventral midbrain cultures showed that the concentrations required for radical scavenging were non-toxic, as reduction in metabolic cell activity or morphological alterations did not occur. 

Negro et al. [77] developed ropinirole (RP)-loaded PLGA MPs as a controlled delivery system for this DA agonist. The formulation exhibited sustained in vitro release of the drug (78.23 µg/day/10 mg MPs) for 19 days. The efficacy of the new delivery system was evaluated in an RT model of PD induced in male Wistar rats. For this, animals received daily i.p. doses of RT (2 mg/kg). Once PD-like symptoms appeared (day 15), animals were given either RP in saline (1 mg/kg/day for 45 days) or RP-loaded PLGA MPs at two dose levels (amounts of MPs equivalent to 7.5 mg/kg or 15 mg/kg RP given on days 15 and 30, respectively). Behavioral outcomes (akinesia, catalepsy, rotarod, swim test) and brain analyses (Nissl staining, glial fibrillary acidic protein (GFAP), TH immunohistochemistry) showed that animals receiving RP either in solution or encapsulated within the MPs reverted PD-like symptoms, with the best results obtained with the MPs at the highest dose assayed.

### 2.4. COMT Inhibitors

Catechol-O-methyltransferase (COMT) is a selective and widely distributed enzyme involved in the catabolism of levodopa, with tolcapone (TC) being a potent COMT inhibitor both in the brain and peripheral tissues, that can slow down levodopa metabolism, thereby leading to a prolongation of its effect in the treatment of PD. Casanova et al. [78] developed TC-loaded PLGA MPs which exhibited zero-order in vitro release of the drug for 30 days. The new delivery system prepared with 120 mg of TC and 400 mg of PLGA 502 was tested in an RT model of PD induced in male Wistar rats. Daily i.p. injections of the neurotoxin (2 mg/kg) were given to induce neurodegeneration. Once established, animals received TC in saline (3 mg/kg/day) or encapsulated within the MPs (amount of MPs equivalent to 3 mg/kg/day TC every 14 days). Brain analyses of Nissl staining, GFAP, and TH as well as behavioral testing (akinesia, catalepsy, swim test) showed that the new delivery system developed for TC was able to efficiently revert PD-like symptoms in the animal model assayed.

### 2.5. α-Synuclein-Targeting Agents

Alpha-synuclein (α-syn) is a key protein involved in the pathogenesis of PD. The exact function of α-syn remains still largely unknown, although mounting evidence supports the fact that α-syn is involved in synaptic plasticity and neurotransmitter release [79,80]. Under normal conditions, native α-syn exists in a dynamic equilibrium between unfolded monomers and α-helically folded tetramers, with a low tendency for aggregation [81]. The reduction in the tetramer:monomer ratio and the consequent increase in the level of α-syn unfolded monomers favor its aggregation [82], which involves a conformational change where the protein adopts a β-sheet-rich structure that facilitates its aggregation into oligomers, protofibrils, and insoluble fibrils that finally accumulate in the form of Lewy bodies [83]. Lewy bodies are a characteristic feature of PD, being closely related to its progression [84], as the appearance of Lewy bodies induces alterations in synapses, mitochondrial dysfunction, and deterioration of the endoplasmic reticulum functionality. These circumstances lead to increased metabolic activity, oxidative stress, and protein accumulation, which may explain the subsequent cell death occurring in the disease. Furthermore, misfolded α-syn can spread between different cells and tissues, inducing misfolding of other units of the protein [83].

The most common α-syn-targeting strategies used in the treatment of PD are decreasing the expression of α-syn with antisense oligonucleotides or miRNA, inhibiting its aggregation with small molecules, favoring its clearance via autophagy, and preventing the seeding and prion-like spreading of α-syn [81,85,86]. In this regard, immunotherapy is gaining increased attention for the management of PD, with active immunization being of particular interest, as it allows for prolonged treatments without the need for frequent dosing intervals [87]. Immunotherapeutic approaches targeting α-syn have the advantage of addressing several of these mechanisms, with active and passive vaccination being able to prevent neurodegeneration and reduce α-syn accumulation by promoting clearance via autophagy and microglial cells [88,89,90,91].

The ability of the combined delivery of antigen plus rapamycin (RAP) in the form of nanoparticles in inducing antigen-specific regulatory T cells (Tregs) has been already demonstrated [92]. Rockenstein et al. [87] adapted this approach to α-syn by developing an antigen-presenting cell-targeting glucan microparticle (GP) vaccine delivery system. This active immunization system with immunomodulatory activity was composed of GP-MPs loaded with RAP and α-syn as antigen to produce the active humoral immunization (GP+RAP/α-syn). The immunity produced was compared to that of GP-alone, GP-α-syn, and GP+RAP in transgenic (tg) PDGFβ α-syn Line D male and female mice. The effect was analyzed using neuropathological and biochemical markers. Mice treated with GP-RAP and GP-RAP/α-syn showed an increase in the number of Treg lymphocytes in the CNS, decreased levels of proinflammatory cytokines (IL-6 and TNF-α), and increased levels of TGF-β1 (transforming growth factor-beta 1). Animals treated with GP-α-syn and GP-RAP/α-syn presented significant increases in anti-α-syn antibody titers, decreased α-syn aggregates and neurodegeneration, and increased functional recovery markers of neurodegeneration (Arc protein and TH). Therefore, combined therapy (RAP and the antigen) led to better responses than monotherapies, possibly due to the combination of the immunomodulatory activity of RAP and immunization by α-syn. The authors concluded that this new vaccine delivery system, which induced both regulatory Tregs and anti-α-synuclein antibody titers, demonstrated a satisfactory effect in reducing α-syn accumulation, neurodegeneration, and inflammation. Authors claimed that this combined vaccine may be more potent than conventional immunization, which is only based on cellular or humoral immunization, with the potential to be further investigated in synucleinopathies, such as PD and dementia with Lewy bodies, among others.

From the promising results obtained, further research should be conducted to determine if combining humoral and cellular immunization might synergistically reduce inflammation and improve microglial-mediated α-syn clearance. 

Table 1 summarizes the polymeric microparticulate systems under investigation as potential new therapeutic approaches for PD.

## 3. Investigational Polymeric Nanoparticles for the Treatment of Parkinson’s Disease

Nanoparticles (NPs) are structures ranging in size from 1 to 100 nm. The biodistribution of NPs is determined by biological barriers. For instance, NPs can cross the BBB, especially when functionalized with surfactants or ligands [93]. This strategy allows for both targeting the drug to the desired compartment and reducing toxicity in other organs and tissues [94,95]. NPs can improve drug solubility, diffusion, and immunogenicity and achieve controlled drug release, with most of these systems being biodegradable and biocompatible [96]. Furthermore, NPs can be administered either systemically or at the site of action. For all these reasons, NPs are currently being extensively investigated as an innovative strategy for the treatment of diseases affecting the CNS, such as PD.

Categories of active ingredients under investigation when encapsulated within polymeric NPs for the treatment of PD include the following.

### 3.1. Neurotrophic Factors

Advances in nanotechnology have been applied to GDNF gene therapy, with promising reports from preclinical studies using DNA nanoparticle gene transfer to achieve long-term GDNF expression [97] and to promote the survival of grafted fetal dopaminergic neurons [98]. Implantation of encapsulated fibroblasts transfected to produce GDNF has also demonstrated behavioral improvements in the 6-OHDA rat model of PD [99]. However, the translational potential of implanting NPs or transfected cells in the human brain is limited by their size, which is larger than the effective pore size of the extra-cellular spaces of the brain. Consequently, limited distribution of these vehicles for GDNF delivery is likely to impair their therapeutic potential. However, recent research efforts are demonstrating that PLGA NPs can protect the encapsulated unstable therapeutic agents from enzyme degradation, deliver the active ingredient in a controlled and continuous mode, and enhance its bioavailability, thereby facilitating drug distribution into the CNS when treating neurodegenerative disorders such as PD [100]. 

Focusing on the application of NTFs (GDNF, VEGF) as therapeutic approaches for preventing the neurodegeneration occurring in PD, Herrán et al. [101] developed PLGA NPs loaded with VEGF or GDNF. Simultaneous addition of VEGF-loaded NPs and GDNF-loaded NPs led to significant protection of PC12 cells against 6-OHDA. In addition, PLGA NPs facilitated the combined delivery of VEGF and GDNF into the brain of 6-OHDA-partially-lesioned male albino Sprague Dawley rats, resulting in continuous and simultaneous delivery of the NTFs. In vivo results led to a significant reduction in the number of amphetamine-induced rotations at the end of the study (10 weeks). Immunohistochemical analysis of TH in the striatum and substantia nigra externa resulted in significant increases of neurons in the VEGF- and GDNF-loaded NP treatment groups, an effect which was found to be synergistic. Therefore, the authors conclude that this synergistic effect may be a valuable neuro-regenerative/neuro-reparative approach for the potential treatment of PD, although further research should be conducted.

The neuroprotective effect of VEFG encapsulated within NPs has been demonstrated after stereotaxic injection but also when given by other administration routes. In this regard, Meng et al. [102] compared the neuroprotective effects of VEGF encapsulated within PLGA NPs injected in the tail vein or directly at the site of action (stereotaxic administration) in male 6-OHDA-lesioned Sprague Dawley rats. The concentrations of NPs evaluated were 1, 10, and 100 ng/mL. The results obtained showed that the greatest neuroprotective effect was produced when the NPs were administered intravenously at the lowest concentration assayed (1 ng/mL). Non-statistically significant differences were found between both modes of administration; however, the intravenous (i.v.) route was simpler and more convenient. Parenteral injection of VEGF-loaded PLGA NPs may be a potentially valuable therapeutic strategy for PD. 

In recent decades, the nasal route has attracted special interest as a convenient, non-invasive, reliable, and safe route to achieve faster arrival and higher drug levels in the brain. Drug delivery by this route allows direct access via the olfactory and trigeminal pathways without the need for crossing the BBB [103] (Figure 5).

BDNF is a potent neuroprotective and neuro-regenerative agent, although its delivery to the brain is limited by poor serum stability and rapid brain clearance. To overcome these limitations, Jiang et al. [100] developed BDNF-loaded poly (ethylene glycol)-b-poly (L-glutamic acid) (PEG-PLE) NPs for intranasal administration. Both PEG and PLE are approved by the FDA, PEG as an inactive ingredient in pharmaceutical preparations and PLE as Generally Recognized as Safe (GRAS). The developed NPs stabilized BDNF, protecting it from unspecific binding to nasal proteins while allowing interaction with its receptors. The neuroprotective effects of the nano-system were evaluated in a Lipopolysaccharide (LPS) acute neuroinflammation model induced in male CD-1 mice. Administration of the nanoformulation improved BDNF access to the brain, especially to the olfactory bulb, hippocampus, and brainstem, compared to non-encapsulated BDNF. The hippocampus and brainstem are important brain regions involved in different CNS disorders. The authors stated that higher brain accumulation of the nano-system could also be related to the observed decreased brain clearance of the NPs as compared with native BDNF. Moreover, NPs increased the half-life of BDNF in the CNS (t_1/2_ = 167 min) with respect to native BDNF (t_1/2_ = 25 min). This observation may help explain why, despite the increased accumulation of BDNF-loaded NPs in the brain, its release into the serum was reduced. Furthermore, intranasally delivered BDNF-loaded NPs produced superior neuroprotective effects in the animal model assayed. The authors concluded that further investigation should be conducted, as the relative surface area of the nasal cavity (surface area/volume) in mice is markedly higher than that in humans, making it difficult to extrapolate the results obtained in the study [104]. 

Nerve growth factor (NGF) is essential for the survival of both peripheral ganglion cells and central cholinergic neurons. In the CNS, this endogenous factor is regulated by neurotransmitters like glutamate and acetylcholine, preventing degeneration of DA neurons [105,106]. For this, it is hypothesized that the administration of NGF might slow down the progression of PD. However, the clinical use of NGF is hampered by its inability to significantly penetrate the BBB. Therefore, the potential clinical applications of NGF as a CNS therapy will depend on the use of suitable carrier systems able to enhance its passage through the BBB. 

Kurakhmaeva et al. [25] developed poly (butyl cyanoacrylate) (PBCA) NPs coated with polysorbate 80, where NGF was adsorbed onto the NPs. The effect of the nano-system was investigated in a parkinsonian syndrome which was induced in C57B1/6 mice by i.p. injection of MPTP. The basic symptoms of the syndrome decreased after administration of the nano-system, as seen from decreased rigidity and increased locomotor activity compared to mice receiving MPTP alone. This effect persisted after 7 and 21 days after single injection of the neurotoxin. In a second stage [107], the authors investigated the effect of i.v. administration of the nano-system in a model of acute scopolamine-induced amnesia in rats, as well as in the MPTP-induced parkinsonian syndrome in C57B1/6 mice. Polysorbate-coated NGF-loaded NPs successfully reversed scopolamine-induced amnesia, also improving recognition and memory. In the MPTP model, the nano-system decreased stiffness and increased locomotor activity as compared to control mice receiving MPTP alone. Furthermore, direct measurement of brain NGF concentrations confirmed the efficient transport of NGF across the BBB. For this, NGF-loaded NPs coated with polysorbate 80 may be an effective carrier system for transporting NGF to the CNS.

The inability of most pharmaceuticals, including NFTs, to cross the BBB is a major challenge for drug development when targeting diseases affecting the CNS. In this regard, polymeric NPs may offer a new approach, although the key issue of potential CNS toxicity has not yet been sufficiently addressed.

Regarding administration routes, studies have shown that intranasal administration of NPs can improve the access and efficacy of NFTs; however, these are preliminary preclinical studies, with further research still needed. 

### 3.2. Natural Antioxidants

Endogenous redox imbalance produced by oxidant and pro-oxidant compounds takes place due to the presence of free radicals, which play a key role in oxidative stress, cell death, and tissue damage. For instance, the formation of ROS is involved in the etiology of PD. Some natural and synthetic drugs with antioxidant properties encapsulated within NPs may be used to strengthen the antioxidant system of the brain [108]. As nicotine has proven neuroprotective against MPTP-induced parkinsonism in animal studies [109], Tiwari et al. [110] developed a new nano-system consisting of nicotine-loaded PLGA NPs. The indicators of oxidative stress, dopaminergic neurodegeneration, and apoptosis were measured both in vitro (culture of dopaminergic neurons from the ventral mesencephalon (VM) region of embryonic day-14 mouse embryos; MPTP neurotoxin) and in vivo (MPTP model of parkinsonism induced in male Swiss albino mice). Administration of the nano-system enhanced bioavailability and brain permeability of nicotine due to sustained release of the drug. In addition, the new formulation allowed cotinine (the active metabolite of nicotine) to be more efficiently available to the cells. The levels of nicotine and cotinine were higher in nicotine-loaded PLGA-NP-treated PD mouse brains compared with the administration of free nicotine. The study demonstrated that the nano-system developed for nicotine resulted in improved neuroprotective efficacy by enhancing its bioavailability, which led to modulation of the indicators of oxidative stress and apoptosis.

Naringenin (NAR) is a flavonoid with high antioxidant activity that has proven efficacy as an experimental therapy for PD. In mice, NAR prevented damage caused by MPTP, preserving motor activity, reducing α-syn aggregation, and maintaining oxidative stress markers at control-like levels [111,112]. However, as with other flavonoids, low bioavailability, poor water solubility, and gastrointestinal degradation limit its potential clinical applications. To overcome these limitations, Md et al. [113] prepared NAR-loaded chitosan NPs for intranasal administration. The results of the cytotoxicity and neuroprotection studies carried out in SH-SY5Y cells, using 6-OHDA as a neurotoxin, showed that NAR-loaded NPs enhanced neuroprotective and antioxidant effects when compared to NAR in solution. Furthermore, permeation studies using sheep nasal mucosa showed that encapsulated NAR exhibited much greater ability to cross the nasal mucosa than NAR in solution. 

Carotenoids are lipophilic antioxidants that could be beneficial in PD therapy due to their ability to quench singlet oxygen, also scavenging other ROS [114,115]. Lutein is a carotenoid present in multiple fruits and vegetables with antioxidant and anti-inflammatory effects, being capable of preventing tissue damage in multiple systems [116,117,118]. Furthermore, lutein has shown protection of dopaminergic neurons in MPTP-treated rats by enhancing the antioxidant defenses and decreasing mitochondrial dysfunction and apoptotic death, which suggests the potential benefits of this carotenoid in PD [116]. Lutein is given orally, but its low solubility and high intestinal/liver first-pass effects result in low oral bioavailability and rapid elimination from the body [119]. To overcome the solubility issue and protect lutein against some external agents such as light, heat, pH, and oxidation, do Prado Silva et al. [119] encapsulated lutein using polyvinyl pyrrolidone (PVP) as a polymer. Free lutein and lutein-loaded PVP NPs were significantly effective in improving cognitive performance in male Swiss mice, with the NPs leading to similar effects at 66 times lower doses that free lutein. Therefore, nanoencapsulation not only enhanced its solubility, but was also able to potentiate lutein activity. 

Fernandes et al. [120] also developed lutein-loaded PVP NPs, which were evaluated in RT-induced parkinsonism in *Drosophila melanogaster* flies. Flies were exposed for 7 days to standard diets containing either RT (500 μM), lutein-loaded PVP NPs (6 μM), or RT (500 μM) and lutein-loaded NPs (6 μM). RT produced locomotor damage, decreased survival rate, and reduced the levels of DA, TH, and acetylcholinesterase (AChE). These effects were reversed by lutein-loaded PVP NPs, providing evidence that the nano-system may be an alternative treatment for RT-induced damage.

Curcumin is a polyphenolic compound isolated from the rhizomes of *Curcuma longa* (turmeric) that exhibits antioxidant, anti-inflammatory, anti-apoptotic, and antimicrobial effects. It is also being investigated as a potential treatment for several brain disorders, including PD [121]. In this pathology, its antioxidant activity and ability to prevent damage induced by MPTP and RT in cell cultures and animal models may be beneficial [122,123]. However, its low water solubility, low oral bioavailability, and rapid elimination from the body limit its potential clinical applications [122]. 

Mogharbel et al. [124] prepared poly (ethylene oxide) (PEO) and poly (ε-caprolactone) (PCL) NPs functionalized with glutathione (GSH) on the outer surface for synergistic delivery of levodopa and curcumin to the CNS, being able to pass the BBB as a potential therapy for PD. Vero and PC12 cells that were treated for up to 72 h with different concentrations of levodopa and curcumin-loaded NPs were blood-compatible and presented low cytotoxicity [124].

Puerarin (PU) is an isoflavone with marked antioxidant activity; however, it exhibits low water solubility and reduced ability to cross the BBB [125]. Puerarin has demonstrated the ability to reduce the toxicity induced by MPTP and RT in cell cultures (PC12 and SH-SY5Y cells) [126,127] as well as in RT-based rodent (male Sprague Dawley rats) models of PD [127]. 

Shiying et al. [128] evaluated the potential neuroprotective role of PU on DA-producing cells both in vitro and in vivo. In vitro, the compound facilitated the differentiation of mesenchymal stem cells (MSCs) to dopaminergic neurons, also promoting cell survival and migration after transplantation of PU-treated DA-producing cells to wild-type male Sprague Dawley rats treated with 6-OHDA.

To improve PU efficacy and its access to the CNS, Chen et al. [125] developed PU-loaded PLGA NPs coated with D-α-tocopherol poly (ethylene glycol) (PEG) 1000 succinate (TPGS), as TPGS has demonstrated to markedly improve the capacity of NPs to cross the BBB [129]. The nano-system was able to prevent MPP+ cytotoxicity and reduce mitochondrial oxidative stress in SH-SY5Y cells to a greater extent than PU in solution. In MPTP-mediated neurotoxicity induced in mice, the nano-system improved disease-associated behavioral deficits and depletion of DA and its metabolites, thereby indicating that PU-loaded NPs may represent a potential therapeutic approach for PD.

Chitosan is a linear biopolymer obtained from the deacetylation of chitin which exhibits nontoxicity, biodegradability, and biocompatibility. It has proven anti-inflammatory, antibacterial, anticancer, and antioxidant properties [130,131]. Chitosan derivatives can be obtained by chemical modifications leading to properties superior to unmodified chitosan being therefore extensively used for the development of drug delivery systems, such as NPs [132]. 

Ahlawat et al. [133] evaluated the antioxidant and anti-apoptotic activities of non-loaded chitosan NPs on a human SH-SY5Y neuroblastoma cell line using RT as a neurotoxin to generate the production of ROS. Cells were exposed to chitosan NPs 6 h before being treated with RT. After 24 h of incubation, differences in cell survival and mitochondrial function were analyzed. The incubation of cells with chitosan NPs reduced RT-induced cytotoxicity and apoptotic cell death, suggesting that chitosan NPs have antioxidant and anti-apoptotic properties. These results suggested that chitosan could be used as a novel therapeutic ingredient, as well as a carrier for combo-therapy. 

Resveratrol (RVT), a phenolic compound found in multiple plant species, has been associated with several beneficial properties, such as cardioprotective, neuroprotective, antioxidant, antitumor, antidiabetic, and antiaging effects, among others [134]. 

With respect to neurodegenerative diseases, including PD, these interesting beneficial properties of RVT may be regulated by several synergistic pathways controlling oxidative stress, cell death, and inflammation [135]. As an antioxidant, RVT has been described to have a double effect: it is said to act as a free radical cleanser, while also being able to increase the activity of antioxidant enzymes [136]. Furthermore, it has been reported that RVT can increase the concentration of some antioxidant enzymes, like glutathione S-transferase, glutathione peroxidase, and glutathione reductase [137]. It has also been proposed that the neuroprotective effects of RVT may be mediated through activation of the PI3-K/Akt signaling pathway, leading to downregulation of the expression of GSK-3β (glycogen synthase kinase 3β) and CREB (cAMP response element-binding protein), which finally resulted in the prevention of neuronal cell death after brain ischemia caused in rats [138]. RVT has demonstrated efficiency in preventing damage to dopaminergic tissue in animal models of PD [139], as the compound exhibits antioxidant activity through ROS uptake, induction of antioxidant enzyme expression, maintenance of mitochondrial function, and anti-apoptotic activity, being also able to promote clearance of α-syn aggregates [135,139,140]. 

However, RVT has poor water solubility and limited stability, being degraded when exposed to pH changes, high temperatures, and certain enzymes. By the oral route, it exhibits low bioavailability and is rapidly eliminated from the body [141]. The use of polymeric NPs may represent an interesting strategy to overcome these limitations. In this regard, da Rocha Lindner et al. [142] developed RVT-loaded poly(lactide) NPs coated with polysorbate 80. The nanoformulation was investigated in C57BL/6 mice receiving for RVT i.p. (nanoparticulate or non-nanoparticulate) for 15 days as well as a single intranasal administration of MPTP. Mice treated with the NPs demonstrated restored behavioral outcomes (olfactory discrimination and social recognition) and exhibited decreased TH levels in the striatum. In this study, polysorbate 80 was used as a surface modifier for the NPs, as it has been shown that this hydrophilic surfactant may facilitate their passage through the BBB [143]. It has been suggested that polysorbate-80-coated NPs adsorb apolipoproteins B and E from the blood after injection, thereby mimicking lipoprotein particles that could be taken up by the brain capillary endothelial cells via receptor-mediated endocytosis. Bound drugs then may be further transported into the brain by diffusion following release within the endothelial cells or by transcytosis [144].

Natural antioxidants seem to be promising agents to prevent or delay the occurrence of PD. Although evidence on the health properties of natural antioxidants is mainly associated with their antioxidant properties, other mechanisms may be involved that still need to be understood. In addition, further research needs to be conducted on the nano-systems developed to elucidate their pharmacodynamics, pharmacokinetics, and side effects. 

### 3.3. MAO Inhibitors

Selegiline (SG), an irreversible and selective MAO-B inhibitor, can alter the metabolic degradation of DA and increase dopaminergic activity by interfering with DA reuptake at the synapse [145]. SG exhibits antioxidant properties independently of its mechanism as an MAO-B inhibitor, as it reduces oxidative radical production, upregulates superoxide dismutase and catalase, and suppresses non-enzymatic and iron-catalyzed auto-oxidation of DA [146]. SG has low bioavailability (approximately 10%), as it undergoes extensive first-pass metabolism after oral administration [147], which limits the amount of drug reaching the CNS. 

Sridhar et al. [148] developed SG-loaded chitosan NPs for intranasal administration to improve the access of the drug to the CNS. The nano-system was evaluated in vivo in both pharmacokinetic and pharmacodynamic studies conducted in an RT experimental model of PD induced in male Sprague Dawley rats. SG concentrations in brain and plasma were 20- and 12-fold higher after intranasal administration in comparison to oral administration. The dose of SG was kept constant at 1 mg/kg to enable pharmacokinetic data comparison. Treatment with intranasal NPs restored DA levels, catalase activity, and glutathione content in the brain, also leading to increased glutathione and significantly decreased lipid peroxidation, which confirmed the antioxidative activity of the drug. Furthermore, better performance in locomotor activity (catalepsy and stride length tests) was obtained. Therefore, intranasally administered SG-loaded NPs resulted in superior therapeutic value when compared to oral administration, thereby showing promise as an approach for the treatment of PD. 

Rasagiline (RG) is another selective and irreversible MAO-B inhibitor. Taking into consideration the bioadhesive characteristics of chitosan to the nasal mucosa and its ability to control the release of active ingredients, together with the interest of incorporating RG in controlled-release systems, Mittal et al. [23] prepared RG-loaded chitosan glutamate NPs. The nano-system was investigated in biodistribution studies in the brain and blood of male Swiss albino mice following intranasal and i.v. administrations. Drug concentrations in the brains after intranasal administration of the NPs were significantly higher at all time points assayed in comparison to both intranasal free drug administration and RG-loaded chitosan NPs given i.v. The results showed a significant improvement in the bioavailability in the brains of animals treated with RG-loaded NPs, which could be a substantial achievement of direct nose-to-brain targeting in PD therapy.

### 3.4. Anti-Inflammatory Agents

Chronic neuroinflammation has been closely related to PD, with evidence for a significant role of the immune system in PD pathogenesis, either through an autoimmune response or inflammation [149], ROS formation, or due to neuronal deterioration [139]. 

The presence of neuroinflammation seems to also play a key role in disease progression [150]. Therefore, anti-inflammatory therapy may be an interesting strategy in the prevention and progression of PD [149,150]. 

Fingolimod, a sphingosine 1-phosphate receptor modulator, is indicated for the treatment of patients with the relapsing–remitting form of multiple sclerosis, due to its immunosuppressive properties in the CNS. Fingolimod has exhibited additional neuroprotective effects [151,152] by stimulating the expression of BDNF and protecting MN9D cells against TNF-α-associated cell death, thereby showing considerable potential for treating synucleinopathies, such as PD [153]. The compound has also demonstrated improved functionality in aging A53T transgenic (Tg) mice, a model that closely resembles PD-like pathology [154]. Fingolimod was also able to revert 6-OHDA- and RT-induced damage in male C57BL/6L mice [155], demonstrating significantly reduced motor function deficits, attenuated decrease in striatal DA and metabolite levels, and diminishment of the loss of TH-positive neurons in the substantia nigra. 

In a study conducted by Ren et al. [156], fingolimod protected against 6-OHDA cytotoxicity and apoptosis in SH-SY5Y cells. In addition, prior administration of the compound to 6-OHDA-lesioned mice ameliorated both nigral dopaminergic neurotoxicity and motor deficits, also reducing neuroinflammation. The authors postulated that these protective effects could be related to activation of AKT (protein kinase B) and ERK1/2 (extracellular regulated protein kinase ½) pro-survival pathways and increased expression of BDNF both in vitro and in vivo, representing a potential candidate for PD therapy. In addition, the compound has proven to specifically reduce microgliosis and astrogliosis, contributing to neuroprotection against 6-OHDA-induced neurotoxicity [157].

Sardoiwala et al. [158] developed fingolimod-loaded chitosan NPs that were evaluated in in vitro and in ex vivo experimental PD models. The neuroprotective activity of the nanoformulation was identified by downregulation of the PD hallmark phospho-serine 129 (pSer129) α-syn and also with antioxidative and anti-inflammatory potential effects. SH-SY5Y cells treated with RT as a neurotoxin were used to evaluate the cytotoxicity of the nanoformulation, resulting in higher neuroprotective capacity against RT when compared to fingolimod in solution. The nano-system was able to regulate the overexpression of nuclear factor kappa B (NF-κB) induced by RT, corroborating its anti-inflammatory potential. Biodistribution tests carried out in BalB/C mice showed that the nanoformulation mainly reached the liver, kidney, spleen, and brain, thereby being efficiently able to cross the BBB. Interestingly, an antioxidant effect of the formulation was also reported, which was attributed to the presence of chitosan, as indicated by other authors [130,133]. This fact could reveal a synergistic effect between fingolimod and chitosan, although more research should be conducted to clarify this potential synergistic action.

Taking into consideration that one of the pathological mechanisms involved in PD is the accumulation of microglia and the fact that when over-activated, microglia can produce cytokines causing inflammation and neuronal death, encapsulating anti-inflammatory agents within polymeric NPs may lead to novel anti-inflammatory therapeutic strategies for neurodegenerative diseases, including PD.

However, treatment of neuroinflammation faces many difficulties due to the poor penetration of drugs across the BBB. In addition, the drawbacks of NPs cannot be ignored, as nanomaterials can induce pro-inflammatory responses, apoptosis, and oxidative stress of neurons in the brain. To avoid this, the use of biodegradable and biocompatible polymers with relatively minimal toxicity could be a potential solution that still needs to be further explored. 

### 3.5. Dopamine Agonists

Delivery of DA to the brain has always been a challenge for scientists when treating PD. In a recent study conducted by Jahansooz et al. [159], the performance of DA-loaded poly (butyl cyanoacrylate) (DA-PBCA) NPs was investigated in vivo. For this, the 6-OHDA model of PD was induced in male Wistar rats. The study demonstrated that the nano-system efficiently restored the loss of DA, also reverting neurodegeneration. In addition, reduced α-synucleinopathy in the animal brains was observed after NP administration. 

Other authors have indicated that the use of DA-loaded chitosan NPs can induce a dose-dependent increase in striatal DA output [160]. Therefore, DA-loaded NPs represent an interesting approach for DA brain delivery and, hence, may be potentially useful for PD therapy.

Monge-Fuentes et al. [161] stated that DA-loaded albumin/PLGA NPs may be a potential innovative therapy for PD due to their ability to improve the access of the drug to the brain. A new formulation consisting of albumin/PLGA NPs loaded with DA (ALNP-DA) was evaluated in 6-OHDA parkinsonism induced in male Swiss albino mice. DA-loaded NPs effectively crossed the BBB, replenishing DA at the nigrostriatal pathway. Significant motor symptom improvement was obtained with the NPs in comparison with animals receiving DA in solution. In addition, ALNP-DA (20 mg/animal) upregulated and restored motor coordination, balance, and sensorimotor performance, leading to outcomes like those present in non-lesioned animals.

Rotigotine (RTG), a non-ergoline dopaminergic agonist, is marketed as a transdermal drug delivery system for the treatment of signs and symptoms of idiopathic PD. The therapeutic potential of oral RTG and its use in long-term therapies is impaired by high first-pass metabolism, low oral bioavailability (1%), and short elimination half-life [162]. 

To improve the access of RTG to the brain, NPs with different ligands have been developed for intranasal administration. In this context, Yan et al. [163] prepared lactoferrin-modified PEG-PLGA NPs loaded with RTG (Lf-RTG-NPs). Lactoferrin (Lf) is a glycoprotein with anti-inflammatory, antimicrobial, and immunomodulatory activities [164]. It has been postulated that Lf accumulates in the brain during neurodegenerative disorders [165]. The Lf receptor is found in the BBB of several species, allowing the transport of Lf through this biological barrier. Therefore, the use of Lf as a ligand has been investigated for improving the access to the CNS of drugs encapsulated within NPs [164]. 

For instance, intranasal administration of Lf-RTG-NPs was evaluated in a 6-OHDA PD model induced in male Sprague Dawley rats. Lf-RTG-NPs provided better biodistribution of RTG in the striatum, improving pharmacodynamic outcomes and neuroprotection over RTG-loaded NPs [163]. 

Bhattamisra et al. [166] prepared RG-loaded chitosan NPs for nose-to-brain delivery. The formulation was tested in cell-based and experimental animal models of PD. Exposure of SH-SY5Y cells to RTG-loaded NPs for 24 h did not cause cytotoxicity. The nanoformulation led to decreased α-syn and increased TH expression in the cells. In addition, the new nanoformulation was evaluated for behavioral and biochemical outcomes in haloperidol-induced PD in Sprague Dawley rats, leading to a reversal of akinesia, catalepsy, and the restoration of swimming ability. Furthermore, a decrease in lactate dehydrogenase (LDH) and an increase in catalase activities were found in the brains of animals treated with the new nano-system, thereby indicating that intranasally-administered NPs enhanced brain-targeting efficiency and RG bioavailability. 

Pramipexol (P) is another dopaminergic agonist with neuroprotective activity that has been encapsulated within NPs for nose-to-brain delivery using chitosan as a polymer [167]. The effects of the nanoformulation were analyzed in RT parkinsonism (3 mg/kg, i.p.) induced in male Sprague Dawley rats. Results from behavioral testing revealed an improved score by photoactometer and reduced catalepsy in animals treated with the NPs in comparison to the administration of the free drug as nasal solution. 

Ropinirole (RP) is a selective dopamine D2 receptor agonist that has been formulated in the form of RP-loaded PLGA NPs as a potential therapeutic strategy for PD [17]. The nano-system was prepared with 8 mg of RP and 50 mg of PLGA 502 and was evaluated in vivo. For this, daily doses of RT (2 mg/kg) were given i.p. to male Wistar rats to induce neuronal and behavioral changes like those present in PD. After 15 days, animals received RP-loaded PLGA NPs (amount of NPs equivalent to 1 mg/kg/day RP every 3 days for 35 days) or RP in saline (1 mg/kg/day for 35 days). Behavioral testing (akinesia, catalepsy, rotarod, swim test) as well as brain histology and immunochemistry (Nissl staining, TH, and GFAP) demonstrated that the NPs developed for RP were able to revert PD-like symptoms of neurodegeneration in the animal model assayed to a greater extent than RP in solution.

From these studies it can be seen that polymeric-nanoparticle-based technologies are promising platforms capable of carrying and releasing dopamine agonists into the CNS; however, assuring that this objective is achieved relies on appropriate composition and physicochemical properties of the delivery systems, which must be carefully chosen to avoid toxicity and ensure successful drug delivery and biological performance.

Several types of nano-systems prepared with different polymers (PLGA, polybutyl cyanoacrylate, chitosan, PEG-PLGA, albumin) have been developed to encapsulate dopamine agonists and tested in PD pre-clinical models; however, to date, no studies have been completed for the treatment of PD at the clinical level.

Although promising results have been obtained both in vitro and in vivo, one important issue remains to be addressed and fully clarified: the toxicity of the nano-systems. 

### 3.6. COMT Inhibitors

Tolcapone (TC) inhibits COMT activity in both the brain and peripheral tissues. It is approved as an adjunctive therapy for PD patients who are treated with levodopa/carbidopa. TC has a short elimination half-life (1.6–3.4 h) [168]. Due to its rapid elimination from the body, the usual dosage schedule of TC is t.i.d. (three times a day). These frequent dosing intervals may have an impact on adherence, leading to less effective treatments for PD.

In this regard, Casanova et al. [78] developed TC-loaded PLGA NPs that exhibited controlled in vitro release of TC for 8 days. The nanoformulation was investigated in vivo in an RT model of PD induced in male Wistar rats. Once neurodegeneration was established, animals received TC in saline (3 mg/kg/day) or encapsulated within NPs (amount of NPs equivalent to 3 mg/kg/day TC every 3 days). Brain analyses of Nissl staining, GFAP, and TH, as well as behavioral testing (catalepsy, akinesia, swim test) demonstrated the ability of the nano-system to efficiently revert PD-like symptoms of neurodegeneration in the animal model assayed.

### 3.7. Other Agents

In this section we include several agents with mechanisms of action still not fully understood in relation to their potential clinical applications in PD. 

Ginkgolide B (GB), a diterpene derivative present in the leaves of *Ginkgo biloba* L., is a potent platelet-activating factor (PAF) receptor antagonist that is also being investigated as a potential neuroprotective agent for CNS disorders, such as PD [169]. GB is known to alter the degradation of TH, the rate-limiting enzyme responsible for DA synthesis, thereby protecting against DA neuron damage induced by 6-OHDA [170] or MPTP [171]. 

Meng et al. [170] demonstrated that GB inhibited PC12 cell apoptosis induced by 6-OHDA in a dose-dependent manner, with this effect being partially mediated by upregulating the Calbindin D28K mRNA and by decreasing intracellular calcium concentrations. Experiments carried out in a reserpine-induced model of PD in male Wistar rats showed that GB ameliorated the reserpine-induced state of oxidative stress, mitochondrial dysfunction, and apoptosis in the brain [171]. However, like many other neuroprotective agents, GB exhibits poor oral bioavailability and cannot readily achieve sufficient exposure in treated patients, thereby limiting its potential clinical application in PD. 

In this regard, Zhao et al. [172] developed a nanoformulation of GB in poly (ethylene glycol)-co-poly(ε-caprolactone) (PEG-PCL). In Madin–Darby canine kidney (MDCK) cells, the NPs were taken up by the cells via multiple nonspecific mechanisms, including clathrin-dependent endocytosis, micropinocytosis, and lipid raft/caveolae-mediated endocytosis. Following oral administration to rats, significant improvement of its oral bioavailability, cerebral accumulation, and bioactivity was obtained, as the nano-system facilitated the sustained release of GB into the blood, thereby improving its ability to reach the brain. In zebrafish, the NPs were readily able to undergo transport across the gastrointestinal, chorion, blood–brain, and blood–retinal barriers. Finally, the authors demonstrated in a PD model induced by MPTP in C57BL/6 mice that the nano-system achieved superior therapeutic efficacy and reduced toxicity than free GB. 

Metformin is a well-known oral antidiabetic that has been identified as a possible treatment for multiple neurological diseases, including PD [173]. However, it is not clear whether its neuroprotective potential is due to an anti-inflammatory effect [174], inhibition of α-syn aggregation [175], or its modulatory effect on the mitochondria [176]. Metformin has an absolute oral bioavailability of 40–60% and rapidly crosses the BBB. In animal studies it has been found that the drug mainly accumulates in the pituitary gland, hippocampus, and frontal cortex, with lower levels found in the striatum, cerebellum, hypothalamus, and olfactory bulb [177]. 

Polyanhydride NPs represent another biodegradable and biocompatible platform for sustained release of therapeutics [178]. Polyanhydrides rich in sebacic acid have shown improved internalization by macrophages, specially through phagocytosis [179]. In addition, one of the mechanisms contributing to dopaminergic neuronal atrophy in PG is mitochondrial dysfunction. In this regard, Schlichtmann et al. [180] developed NPs from 1,6-bis (p-carboxyphenoxy) hexane and sebacic acid loaded with mito-metformin (a conjugate of metformin and TPP with higher affinity for mitochondria) that was functionalized with (3-carboxypropyl) triphenyl-phosphonium (CPTP). The nanoformulation was tested in an RT-induced PD model using N27 cells, a rat mesencephalic neuronal cell line. The results obtained showed improved cellular uptake, which ameliorated the RT-induced mitochondrial dysfunction. This outcome was not observed with nonfunctionalized NPs or mito-metformin in solution. The authors stated that the targeted NPs platform may provide enhanced dose-sparing and potentially reduce the appearance of systemic side effects in PD treatment. 

Finally, retinoic acid (RA) is a metabolite of retinol that plays an important role in the development of the mammalian nervous system and in the maintenance of the nigrostriatal pathway [181]. Its ability to reduce the susceptibility of neuronal cells to toxins such as MPTP and 6-OHDA has been described [182,183]. Moreover, it has been postulated that RA can reduce neuroinflammation [184] and alter the autophagy of macromolecules [185], although its mechanism of action is not yet fully understood. On the other hand, RA has low water solubility, a short elimination half-life, and narrow therapeutic index. For these reasons, Esteves et al. [181] studied the neuroprotective effects of RA-loaded polyethylenimine and dextran sulfate NPs in an MPTP model of PD induced in C57BL6 mice. Intrastriatal injections of the nano-system (100 ng/mL RA-NPs) reduced the loss of TH+ cells and TH protein levels in the ipsilateral substantia nigra to levels similar to control animals. The nanoformulation led to a reduction in the destruction of dopaminergic tissue both in the striatum and substantia nigra. In addition, increases of the transcriptional factors Pitx3 and Nurr1, both involved in the survival of dopaminergic tissue, were found, thereby indicating the potential interest of this new formulation as an efficient and suitable strategy to prevent the onset of PD.

Ginkgolide B, mito-metformin, and retinoic acid have been encapsulated within different types of polymeric nano-systems, with promising results obtained both in cell cultures and in animal models of PD.

Although current research shows promise for these nano-systems in PD, there are still concerns that need to be highlighted, and they apply not only to research into the drugs, but to the nano-systems as well. For instance, in the studies reviewed, toxicity issues were not sufficiently addressed. In the case of NPs, formulation parameters such as particle size, shape, surface, and charge should be taken into consideration. Moreover, when developing nano-systems for PD or other brain disorders, it is not enough for the NPs to effectively cross the BBB; they must also preserve the pharmacological activity of the encapsulated agent without causing toxicity. 

Table 2 summarizes the polymeric nanoparticulate systems under investigation as potential new therapeutic approaches for PD.

## 4. Discussion

Multiparticle systems comprise a wide variety of drug delivery systems with many different applications. In this review, we have focused specifically in polymeric micro- and nano-systems developed for the treatment of PD. 

The interest of encapsulating antiparkinsonian agents within polymeric micro- and nano-systems has resulted in increasing research efforts over the past 20 years into their potential as therapeutic strategies for PD, a devastating neurodegenerative disease. 

In this context, different types of the antiparkinsonian agents are under preclinical research, including neurotrophic factors, natural antioxidants, MAO inhibitors, COMT inhibitors, dopamine agonists, and anti-inflammatory agents, among others, with PLGA being the polymer most frequently used for the development of micro- and nano-systems for PD, as it is approved by the FDA and the EMA for use in humans due to its properties of biocompatibility and biodegradability, suitable mechanical properties, and biodegradation kinetics, as well as ease of processing. Chitosan is also frequently used for the encapsulation of therapeutic agents destined to treat PD, as it is a biodegradable, biocompatible polymer regarded as safe for human dietary use and approved for wound dressing applications (Figure 6).

When preparing multiparticulate drug delivery systems, a wide range of variables must be considered, each of which can have a decisive effect on the final properties of the device and consequently, on the effect obtained in vivo. Solubility characteristics of the therapeutic agent, the polymer or combination of polymers selected, the percentage of each polymer and active ingredient, solvents and cosolvent, the method of preparation, etc., are factors that can lead to different properties and behavior of the final delivery system [186].

Micro- and nano-systems exhibit different features, not only in terms of size, but also regarding their physicochemical properties, with the success of a device in the form of polymeric microparticles not necessarily reached with the same therapeutic agent and polymer when encapsulating as nanoparticles, and vice versa [78]. One of the main differences is related to their ability in crossing biological barriers, especially the BBB, which is needed when targeting neurodegenerative diseases, such as PD. The CNS is sequestered from the general circulation by the tight endothelium of the BBB, which allows the passage of small lipid-soluble molecules while limiting the access of pathogens or toxins. Microparticles cannot cross the BBB, whereas evidence on the penetration capacity of nanoparticles, especially functionalized nanoparticles, is vast and diverse [187]. 

PD is the second most common neurodegenerative disorder after Alzheimer’s disease, with its incidence constantly growing due to increased life expectancy globally. Currently more than 10 million people worldwide are living with PD. Projection analysis suggest that the overall PD incidence will continue to increase in the coming decades. 

In this regard, research efforts are needed to find new therapeutic agents and/or improve the efficacy of the existing ones. For this, the development of polymeric micro- and nano-systems is an interesting strategy in which, as reviewed in this work, many researchers are involved. 

From this review, it can be stated that research conducted on antiparkinsonian agents encapsulated within polymeric micro- and nano-systems, although extensive, is still in the preclinical stages of both in vitro and in vivo research, with no clinical trials found in the literature to date. Although extensive research has been carried out, as discussed in this review, to date none of the micro- or nano-systems have reached the market, not even the clinical stages of research [188]. Extrapolating in vitro (cell cultures) and in vivo results obtained in experimental animals to humans is a key factor that conditions and determines the potential fate of the delivery systems developed, as the relevance of results obtained from simplified systems to humans may be questionable. 

Cell cultures and animal models cannot precisely replicate human pathophysiology, with the limited efficacy demonstrated by the existing drugs when treating PD as an indication of its complex etiology and pathogenesis.

In addition, in the case of nanoparticles, the risk caused by the systemic accumulation of particles in the human body cannot be ignored. Another limiting factor is related to the laws and regulations of the countries regarding the use of nanomedicines. In general, there is a lack of unified global regulatory criteria for the use of nanomaterials and nanomedicines [189].

The leap from preclinical stages of research to the approval of new medicines is very large, mainly due to the fact that in many cases, the results in humans are not those potentially expected from the results obtained in preclinical research. 

However, despite all these challenges, the unique properties of polymeric micro- and nano-systems, such as targeting the site of action, reducing side effects, biocompatibility, and providing sustained and controlled drug release patterns, make them very promising therapeutic strategies in which many research efforts are currently being made. 

In this regard, Table 3 summarizes the latest patent applications found in the field of polymeric MPs for PD. Patents for polymeric NPs destined to treat PD have not been found. 

From the review conducted in this work, many of the delivery systems have demonstrated beneficial effects both in vitro and in experimental animal models of PD, thereby representing potentially interesting strategies for the delivery of either already approved drugs or new antiparkinsonian agents. 

## 5. Conclusions

To date, there are several drug delivery devices in the form of micro- and nano-systems approved for clinical use by the FDA. Most of them are microparticulate systems, with just a few in the form of nanoparticles, which are mainly destined to treat different types of cancer. However, to date, no such system is found on the market, not even in clinical trials for treating PD. Nevertheless, as reviewed in this work, many polymeric micro- and nano-systems are currently under preclinical investigation as new therapeutical approaches for PD, a devastating neurodegenerative disorder that still requires great research efforts.

## Figures and Tables

**Figure 1 pharmaceutics-15-00013-f001:**
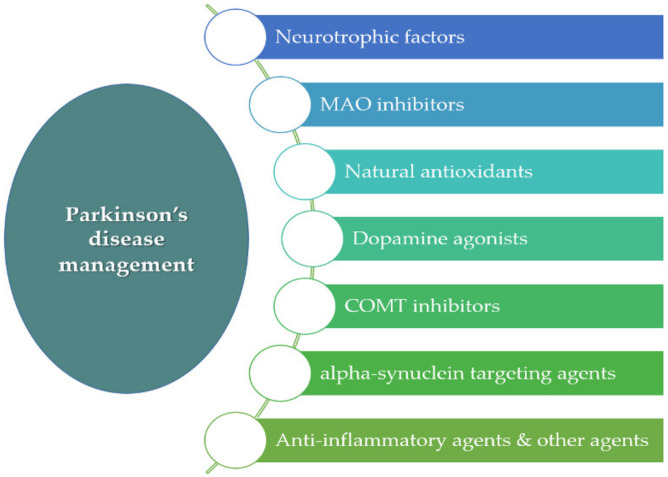
Antiparkinsonian agents under investigation encapsulated within polymeric micro- and nano-systems.

**Figure 2 pharmaceutics-15-00013-f002:**
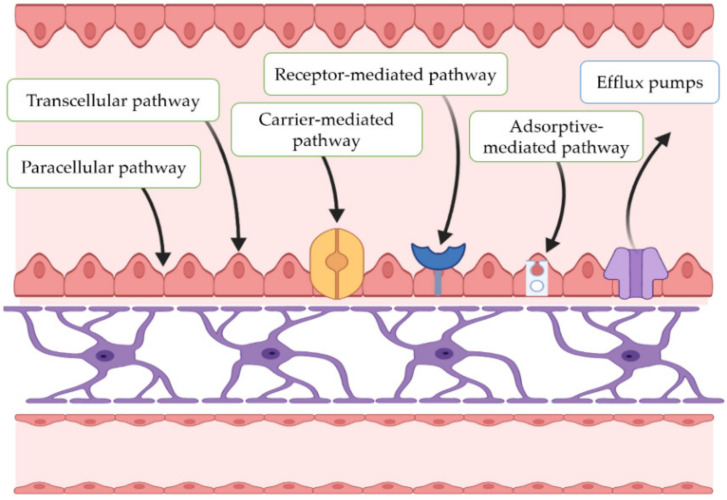
Possible pathways to cross the blood–brain barrier (BBB).

**Figure 3 pharmaceutics-15-00013-f003:**
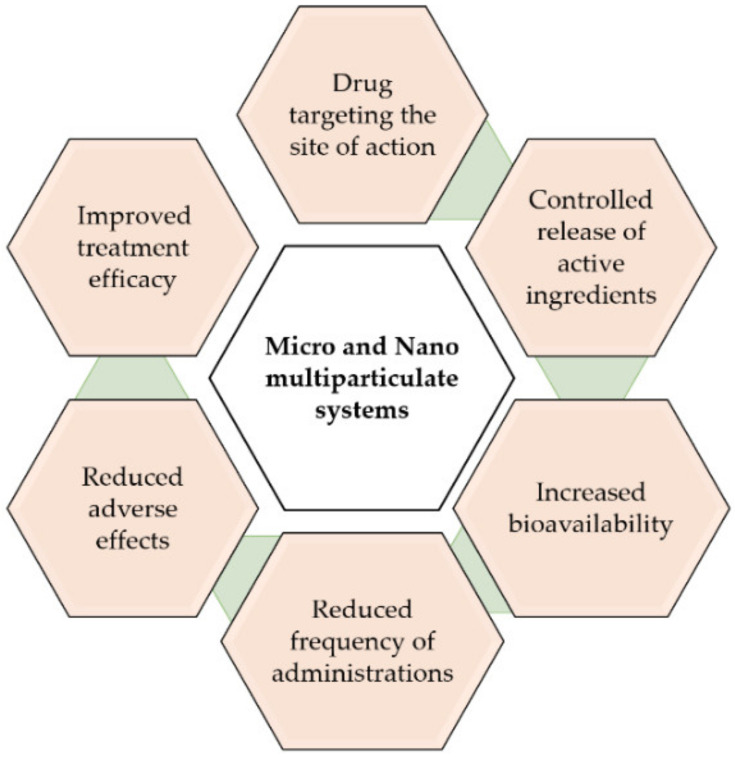
Advantages of multiparticulate drug delivery systems compared to conventional medications.

**Figure 4 pharmaceutics-15-00013-f004:**
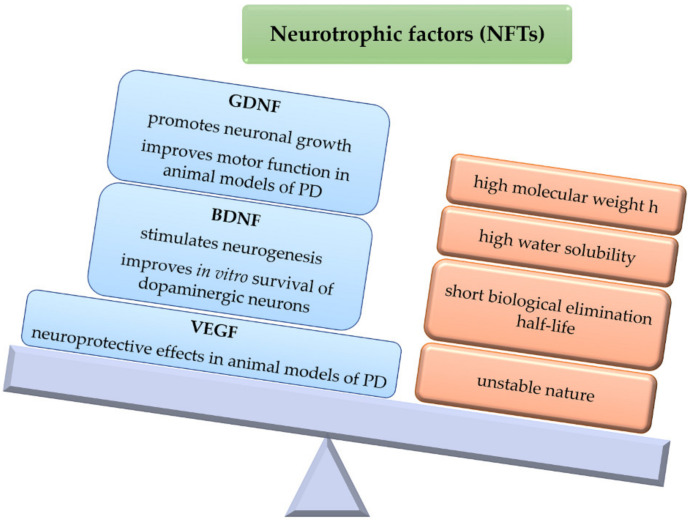
Neurotrophic factors. Potential mechanisms of action in Parkinson’s disease (PD). Physicochemical and biological characteristics. GDNF (Glial-Cell-Derived Neurotrophic Factor), BDNF (Brain-Derived Neurotrophic Factor), VEGF (Vascular Endothelial Growth Factor).

**Figure 5 pharmaceutics-15-00013-f005:**
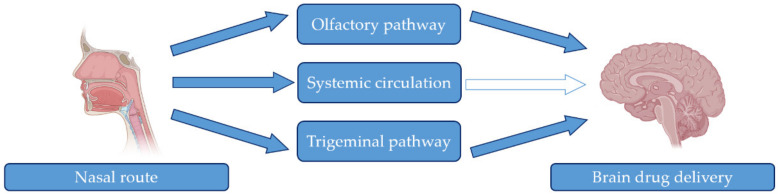
Nose-to-brain drug delivery.

**Figure 6 pharmaceutics-15-00013-f006:**
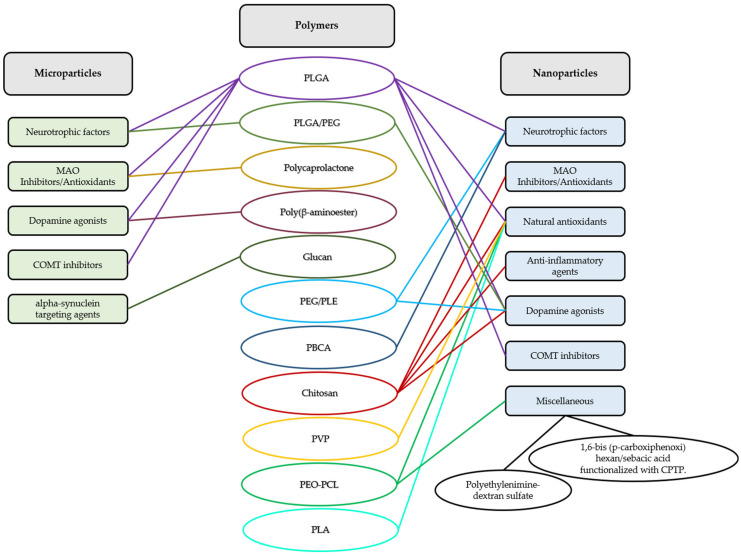
Polymers used for the encapsulation of antiparkinsonian agents within investigational micro- and nano-systems.

**Table 1 pharmaceutics-15-00013-t001:** Polymeric microparticulate systems under investigation for the treatment of PD.

Category	Active Compound	Polymer	Research Model	Ref.
Neurotrophic factors	GDNF	PLGA shell/Collagen core	Cell culture: neural stem/progenitor cells.	[44]
	GDNF/BDNF	PLGA/PEG	Cell cultures: brain/glial cells, microglia, and astrocytes. Animal study: female Sprague Dawley rats.	[43]
	GDNF	PLGA	Animal study: partial and progressive 6-OHDA-induced model of PD. Female Sprague Dawley rats.	[42]
	GDNF	PLGA	Animal study: 6-OHDA-induced model of PD. Female Sprague Dawley rats.	[46]
	GDNF	PLGA	Animal study: MPTP-induced model of PD. *Macaca fascicularis*.	[47]
	GDNF/VEGF	PLGA	Animal study: 6-OHDA-induced model of PD. Female albino Sprague Dawley rats.	[40]
MAO inhibitors/Antioxidants	Rasagiline	PLGA	SKN-AS cell culture, H_2_O_2_ neurotoxin. Animal study: RT-induced model of PD. Male Wistar rats.	[69]
	Rasagiline	PLGA	SKN-AS cell culture, H_2_O_2_ neurotoxin. Animal study: advanced RT model of PD. Male Wistar rats.	[70]
	Rasagiline	Polycaprolactone	Animal study: RT-induced model of PD. Male Sprague Dawley rats.	[22]
Dopamine agonists	Dopamine	Poly(β-aminoester)	Cell cultures: SH-SY5Y cells, rat primary astrocytes, embryonic midbrain cultures.	[76]
	Ropinirole	PLGA	SKN-AS cell culture, H_2_O_2_ neurotoxin. Animal study: RT-induced model of PD. Male Wistar rats.	[77]
COMT inhibitors	Tolcapone	PLGA	Animal study: RT-induced model of PD. Male Wistar rats.	[78]
Alpha-synuclein-targeting agents	α-syn + Rapamycin	Glucan	Animal study: transgenic PDGFβ α-syn. Male and female Line D mice.	[87]

**Table 2 pharmaceutics-15-00013-t002:** Polymeric nanoparticulate systems under investigation for the treatment of PD.

Category	Active Compound	Polymer	Research Model	Ref.
Neurotrophic factors	GDNF/VEGF	PLGA	Animal study: 6-OHDA-induced model of PD. Male Sprague Dawley rats.	[101]
	VEGF	PLGA	Animal study: 6-OHDA-induced model of PD. Male Sprague Dawley rats.	[102]
	BDNF	PEG-PLE	Animal study: LPS-acute neuroinflammation model. Male CD-1 mice.	[100]
	NGF	PBCA coated with polysorbate 80	Animal study: MPTP-induced model of PD. C57B1/6 mice.	[107]
Antioxidants	Nicotine	PLGA	Cell culture: dopaminergic neurons from the ventral mesencephalon region of mouse embryos. Animal study: MPTP-induced model of PD. Swiss albino mice.	[110]
	Naringenin	Chitosan	Cell culture: SH-SY5Y cell line, 6-OHDA neurotoxin.	[113]
	Lutein	PVP	Adult male Swiss mice.	[119]
	Lutein	PVP	Animal study: RT-induced parkinsonism. *Drosophila melanogaster.*	[120]
	Curcumin/L-DOPA	PEO-PCL coated with GSH	Cell culture: Vero and PC12 cells.	[124]
	Puerarin	PLGA coated with D-α-tocopherol PEG 1000 succinate	Cell culture: SH-SY5Y cell line, MPP^+^ neurotoxin. Animal study: MPTP-induced model of PD. Sprague Dawley rats.	[125]
	Chitosan	None	SH-SY5Y cell line, RT neurotoxin.	[133]
	Resveratrol	PLA	Animal study: MPTP-induced model of PD. C57BL/6 mice.	[142]
MAO inhibitors/Antioxidants	Selegiline	Chitosan	Animal study: RT-induced model of PD. Male Sprague Dawley rats.	[148]
	Rasagiline	Chitosan	Animal study: male Swiss albino mice.	[23]
Anti-inflammatory agents	Fingolimod	Chitosan	Cell culture: SH-SY5Y cell line, RT neurotoxin. Animal study: BalB/C mice.	[158]
Dopamine agonists	Dopamine	PBCA	Animal study: 6-OHDA-induced model of PD. Male Wistar rats.	[159]
	Dopamine	albumin/PLGA	Animal study: 6-OHDA-induced model of PD. Male Swiss albino mice.	[161]
	Rotigotine	PEG-PLGA	Animal study: 6-OHDA-induced model of PD. Male Sprague Dawley rats.	[163]
	Rotigotine	Chitosan	Cell culture: SH-SY5Y cell line. Animal study: Haloperidol-induced model of PD. Sprague Dawley rats.	[166]
	Pramipexol	Chitosan	Animal study: RT-induced model of PD. Male Sprague Dawley rats.	[167]
	Ropinirole	PLGA	Animal study: RT-induced model of PD. Male Wistar rats.	[17]
COMT inhibitors	Tolcapone	PLGA	Animal study: RT-induced model of PD. Male Wistar rats.	[78]
Other agents	Ginkgolide B	PEG-PCL	Cell culture: Madin–Darby canine kidney (MDCK) cells. Animal studies: zebrafish. MPTP-induced model of PD in C57BL/6 mice.	[172]
	Mito-metformin	1,6-bis (p-carboxiphenoxi) hexan/sebacic acid functionalized with CPTP.	Cell culture: N27 cell line, RT neurotoxin.	[180]
	Retinoic acid	Polyethylenimine-dextran sulfate	Animal study: MPTP-induced model of PD. C57BL6 mice.	[181]

**Table 3 pharmaceutics-15-00013-t003:** Patents related to polymeric microparticulate systems for PD filed recently [190].

Patent/Application Number	Patent Title	Assignee	Polymers	Publication Date
CN200910201414.X	Selegiline sustained release microparticles and method for preparing same	Suzhou University (Jiangsu, China)	PLGA/PLA	19 May 2010
CN01884623-B	Levodopa methyl ester slow-release microsphere composition and preparation method thereof	Xinhua Hospital Affiliated to Shanghai Jiaotong University School of Medicine (Shanghai, China)	Degradable hydrophobic polymers	20 July 2010
WO2012009973-A1	Antiparkinsonian drug-loaded microsphere composition and use thereof	Xinhua Hospital Affiliated to Shanghai Jiaotong University School of Medicine (Shanghai, China)	PLGA, PLA, polycaprolactone, or their mixtures	16 February 2011
WO2012068783	Compositions of rotigotine, derivatives thereof, or pharmaceutical acceptable salts of rotigotine or its derivative	Shandong Luye Pharmaceutical Co. Ltd. (Shanghai, China)	PLGA	31 May 2012
WO2018223895	Long-acting sustained-release preparation for resisting Parkinson’s disease and preparation method thereof	AC Pharmaceuticals Co., Ltd. (Kolkata, India)	Biodegradable high-biocompatibility molecular polymers	13 December 2018
WO2020105883	Method for manufacturing rotigotine-containing polymer microparticles	Chemtech Research Incorporation and EWHA University Industry Collaboration Foundation (Hwaseong-si, Korea)	hydrophobic polyester-based polymer polylactic acid (PLA), poly (lactic acid-co-glycolic acid) (PLGA), and polylactide	28 May 2020
CN202011222237.6	Rasagiline mesylate microsphere preparation and preparation method thereof	China Pharmaceutical University (Jiangsu, China)	PLA	8 January 2021
CN200910201416.9	Rivastigmine slow-release micropheres and preparation method thereof	Suzhou University (Jiangsu, China)	PLGA/PLA	21 January 2021
WO2021207737	Long acting apomorphine formulations and injectors for therapeutic delivery of the same	Scienture, Inc. (Hauppauge, USA)	Biodegradable polymers	14 October 2021

## Data Availability

Not applicable.

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
