# Peer review of "Antiparkinsonian Agents in Investigational Polymeric Micro- and Nano-Systems"

_pharmaceutics, 2022, doi:10.3390/pharmaceutics15010013_

Round 1
Reviewer 1 Report
In this review, Paccione and co-workers summarized the use of polymeric nanoparticles and microparticles as the drug delivery systems for Parkinson's disease treatment. Frankly speaking, this review article needs substantial improvements prior to its publication in any journals. Some major comments are listed below for your consideration.
1. Throughout the whole review artile, there is no figure or scheme but only two roughly-prepared tables to show us limited information about the formulation of the polymeric particulate systems for PD treatment. One picture is equal to one thousand words. I would suggest the authors to take more time to prepare and include at least five to ten figures or schemes, which would significantly improve the readability of the review.
2. Section 1 and Section 2 are basically described in the same way based on the category of active ingredients encapsulated within the polymeric particulate systems.However, according to the title and abstract of this review, the major focus of this paper is supposed to be the polymeric delivery systems based upon nanoparticles and microparticles, but not the active ingredients. I would suggest the authors to re-organize the manuscript following a more logic line.
3. Comparison of polymeric nanoparticles and microparticles for drug delivery in PD treatment is of fundamental importance in this area; however, the related assessment and discussion are missing in this review.
4. The conclusion of this revie article needs re-writing. I would suggest the authors to in-depth rethink about the current trend and future development of this research area and provide more insightful opinions, thus bringing new research opportunities to the field.
Author Response
Point 1:
Answer: Following the indications given by the reviewer six new figures have been included in the manuscript (see pages 2, 3, 4, 12, and 22).
Point 2:
Answer: We agree with the reviewer in that the original title of the article does not reflect its content. As this concern was also raised by another reviewer, the title has been changed to match the structure of the paper more accurately.
Point 3:
Answer: The requested content has been added in the discussion section of the manuscript (see discussion section).
Point 4:
Answer: The discussion and conclusion sections of the manuscript have been modified (see discussion and conclusion sections).
Reviewer 2 Report
This is an interesting review on application of polymeric micro- and nano- carriers of presumable agents against Parkinson's disease. Paper is comprehensive, well structured and well written. Despite of this ist is difficult to follow because covers quite huge material. In this respect the Tables are good summary of the text. There are some very small errors, which can be corrected upon proof-reading.
They are as follows:
1./ explanation what are Tregs should be in line 363 not 371;
2./ It is usually said that nanoparticles size is up to 100 nm (line 385)
3./ names of organisms should be in italics (lines 538 & 780).
Author Response
Point1:
Answer: Explanation of Tregs has been included in the text (see line 384).
Poine2:
Answer: Nanoparticles size has been corrected (see line 415).
Point3:
Answer: Organisms names have been changed to Italics (see lines 567 and 813).
Reviewer 3 Report
The current manuscript provides a not-so-novel account of nano and micro strategies for Parkinson's disease intervention. I have following concerns about the manuscript:
1. The title of the manuscript is misleading as the strategies mentioned by the authors are not new... the research may be new but not the approach.
2. There are paragraph after paragraph describing the studies in much detail. But the authors did not put much effort into creating a narrative and providing a critical overview and future strategies.
3. The another concern is with the clinical and commercial part of the review. How many such strategies are patented? How many of them are undergoing clinical trials? Are there any such products in the market?
Author Response
Point 1:
Answer: Following the indications given by the reviewer the title of the manuscript has been changed.
Point 2:
Answer: Discussion sections have been added to the manuscript and conclusion section have been modified.
Point 3:
Answer: Concerns raised by the reviewer have been addressed in the new discussion and conclusion sections of the manuscript. From the review performed in this work to date all micro- and nano-systems for PD are in the stage of preclinical research with none of them undergoing clinical experimentation. None of the reviewed strategies are in the market.
Round 2
Reviewer 1 Report
I have no further comment or question now. I hope this review will be useful for researchers in the field.
Author Response
We truly appreciate your thoughtful comments that helped us to improve our manuscript.
Reviewer 3 Report
These two comments are still not addressed. The revised manuscript do not show such revisions.
2. There are paragraph after paragraph describing the studies in much detail. But the authors did not put much effort into creating a narrative and providing a critical overview and future strategies.
3. The another concern is with the clinical and commercial part of the review. How many such strategies are patented? How many of them are undergoing clinical trials? Are there any such products in the market?
Author Response
Point 2:
Answer: questions raised by the reviewer have been addressed in the revised version of the manuscript (see lines 233-239, 312-316, 423-425, 538-544, 679-684, 761-771, 837-847, 931-941, 1002-1004).
Point 3:
Answer: this question has been addressed in the revised version of the paper: see discussion and conclusion sections (lines 993-998 in the discussion section and lines 1031-1032 in the conclusion section). Also, a new reference has been added from the search for clinical trials carried out (see now ref. 188). Regarding patents a new table (see table 3) has been included in the text.
Round 3
Reviewer 3 Report
No further comments.
Author Response

(The authors gave the same response as above.)
